# Risk factors for metabolic bone disease of prematurity: A meta-analysis

Jie Wang[1‡], Qian Zhao[1‡], Baochang Chen[1], Jingfei Sun[2], Jiayu Huang[1], Jinfeng Meng[1], Shangbin Li[1], Weichen Yan[1], Changjun Ren[1]*, Ling Hao[1]*

1 The First Hospital of Hebei Medical University, Shijiazhuang, Hebei Province, China, 2 People's Hospital of Zhengding County, Shijiazhuang, Hebei Province, China

‡ JW and QZ are contributed equally and first coauthors to this work.
* 137544907@qq.com (CR); 2475759137@qq.com (LH)

## Abstract

### Objective

To investigate the risk factors for metabolic bone disease of prematurity (MBDP), and to provide a reference for the prevention of MBDP.

### Methods

The databases including China Biomedical Literature Service System, China National Knowledge Infrastructure, Wanfang Data, and Weipu Periodical Database, PubMed, Web of Science, Embase, Cochrane Library and other databases were searched for studies on the risk factors for MBDP published up to June 18, 2021. RevMan 5.3 and Stata 14.1 software were used to perform a Meta analysis.

### Results

A total of 15 articles were included, including 13 case-control studies, 1 current investigation, and 1 retrospective cohort study. There were 1,435 cases in the case group and 2,057 cases in the control group, with a total sample size of 3,492 cases. Meta analysis showed that risk factors for MBDP include birth weight <1000g (OR = 6.62, 95%CI: 2.28–19.25), gestational age <32 weeks (OR = 2.73, 95%CI: 1.07–6.95), septicemia (OR = 2.53, 95%CI: 1.69–3.79), parenteral nutrition time (OR = 4.04, 95%CI: 1.72–9.49), cholestasis (OR = 3.50, 95%CI: 1.49–8.23), intrauterine growth retardation (OR = 6.89, 95%CI: 3.81–12.44), while the birth weight(OR = 0.44, 95%CI: 0.21–0.90) and gestational age (OR = 0.57, 95% CI: 0.44–0.73)are the protective factors of MBDP.

### Conclusion

Factors like birth weight <1000g, gestational age <32 weeks, septicemia, parenteral nutrition time, cholestasis, and intrauterine growth retardation may increase the risk of metabolic bone disease of prematurity.

**Data Availability Statement:** All relevant data are within the paper and its Supporting Information files.

**Funding:** The authors received no specific funding for this work.

**Competing interests:** The authors have declared that no competing interests exist.

# 1 Introduction

Metabolic bone disease of prematurity (MBDP), characterized by a decrease in bone-like tissue and bone mineral content and possible biochemical changes in calcium and phosphorus metabolism, is a multifactorial systemic disease affected by nutritional and biomechanical factors. The essence is that the bone minerals of preterm infants is not sufficient for normal bone growth and development, which can be accompanied by blood biochemical and imaging changes, such as hypophosphatemia, hyperalkaline phosphatase, bone mineralization deficiency and other imaging manifestations [1]. Previous studies show that the incidence of MBDP in very low birth weight (VLBW) and extremely low birth weight (ELBW) preterm infants are respectively 32% and 54% [2].

The diagnostic criteria for metabolic bone disease of prematurity are not uniform, and the diagnosis of MBDP requires a comprehensive review on medical history, clinical manifestations, biochemical indicators and imaging tests [3, 4]. MBDP has an insidious onset and is asymptomatic in the early stages until severe bone demineralization occurs. The most obvious clinical manifestations are cranial deformities, including enlarged cranial sutures, enlarged anterior fontanelle, forehead bulge and cranial softening, thickening of the rib and rib cartilage junction and carpal joints, and rib or long bone fractures for severely patients [5]. Neonatal bone quality is evaluated by biochemical indicators and imaging tests. The most commonly used blood biochemical indicators are serum calcium, serum phosphorus, alkaline phosphatase (ALP), parathyroid hormone (PTH) and 25hydroxyvitamin D (25(OH)D). Blood calcium levels in the body are regulated by both calcitonin and parathyroid hormone. When blood calcium decreases, the body maintains blood calcium levels by mobilizing bone calcium under the regulation of parathyroid hormone. Blood calcium can be normal or high when the body is deficient in calcium, and it only decreases when bone calcium reserves are depleted in the late stage of MBDP. Therefore, diagnosing MBDP in the early stage with blood calcium is meaningless. The earliest blood biochemical changes in infants with MBDP are characterized by hypophosphatemia. Blood phosphorus concentrations are a good indicator to review bone phosphorus reserve, and a on-going decrease in blood phosphorus suggests inadequate phosphorus intake and an increased risk of osteoporosis. When hypophosphatemia persists, bone resorption increases, calcium excretion via the kidneys continues to increase, and a state of calcium depletion ensues. Increased blood alkaline phosphatase levels are associated with the development of MBDP, and increased blood alkaline phosphatase levels can precede the onset of clinical symptoms. The secretion of PTH is mainly regulated by plasma calcium ion concentration. The blood calcium level is maintained by mobilizing osteolysis, promoting calcium reabsorption by the renal tubules, and phosphate excretion. A UK survey found that plasma parathyroid hormone is used as a supplementation tool to guide neonatologists in MBDP screening, diagnose and monitoring, yet is underutilized to fully play its role [6]. The main etiology of MBDP is calcium and phosphorus deficiency, while serum 25(OH)D can be normal, decreased or even increased, so 25(OH)D is not used as a diagnostic indicator of MBDP. Urinary biochemical indicators include urinary calcium, urinary phosphorus, urinary calcium/creatinine, urinary phosphorus/creatinine and tubular reabsorption of phosphorus (TRP). Increased urinary calcium and urinary phosphorus suggest better bone mineral deposition. Imaging tests is to measure bone mineral density, mainly by X-ray and dual energy X-ray absorptiometry (DEXA). X-rays of MBDP may show osteoporosis at the ends of long bones, cupping or burr-like changes at the epiphysis, enlarged rib ends, subperiosteal new bone formation or fractures. X-rays are only suitable for the diagnosis of severe MBDP with significant osteoporosis or bone fractures, for it may not discover osteoporosis with <20%-40% bone loss [7]. Therefore, although X-rays are highly specific for the diagnosis of MBDP, they are not

suitable for early diagnosis. DEXA, on the other hand, is the gold standard for the diagnosis of osteoporosis, reflecting the two-dimensional area density of the bone, but not the three-dimensional density of the bone. The use of DEXA for screening of MBDP is technically difficult and not suitable for routine screening. At present, the diagnosis of MBDP is mostly based on typical clinical manifestations and radiographic findings, but by that time the bone mineral density may have significantly decreased. Since most MBDP has no obvious clinical symptoms, its diagnosis is mainly based on early clinical screening and monitoring.

Prevention is more important than treatment, for metabolic bone disease of prematurity, and the focus of bone health management in preterm infants is to provide adequate calcium and phosphorus intake to promote normal bone growth [4]. Postnatal calcium and phosphorus absorption rates in preterm infants are positively correlated with age in days, calcium, phosphorus, lactose and intake in fat, and are also influenced by vitamin D levels. In clinical practice, preventive measures should be implemented for preterm infants with high-risk factors, and nutritional management, especially calcium, phosphorus and vitamin D intake, should be strengthened for very low birth weight infants. Prolonged use of drugs affecting bone metabolism should be limited, and biochemical indicators should be actively monitored. After discharge, infants at high risk of MBDP should continue to be fed with nutritional formula until correction at full term or until there is no evidence of combined MBDP on regular clinical monitoring. Infants at risk for MBDP may be trained in daily passive exercises to prevent MBDP after achieving total enteral feeding, and if diagnosed with MBDP, comprehensive nutritional management measures should be promptly implemented [4]. The key to treatment is supplementation with calcium, phosphorus and vitamin D preparations on the basis of intensive nutritional formula feeding, eunsuring it reaches the target amount so to correct abnormal metabolic states such as hypophosphatemia, secondary hyperparathyroidism and vitamin D deficiency as soon as possible. Supplementation of phosphorus preparations alone can aggravate the imbalance of calcium and phosphorus in the body, leading to secondary hyperparathyroidism and aggravating bone lesions. Therefore, it is emphasized that infants with MBDP should be given additional calcium and phosphorus supplementation on top of strengthened formula feeding. Infants with MBDP are in need of concomitant vitamin D supplementation to promote intestinal absorption of calcium and phosphorus. Improvement can be seen in imaging results after several weeks with increased enteral or parenteral mineral supplementation. The efficacy of treatment can be assessed by imaging when treatment reaches 6–8 weeks. The prognosis of MBDP is influenced by a number of factors, which has not been clarified. To reduce MBDP complications and improve their short- and long-term prognosis and linear growth, regular follow-up and monitoring are emphasized in preterm infants with MBDP risk factors. The goal is to maintain normal blood calcium and phosphorus, avoiding excessive urinary calcium excretion; and to maintain the desired growth in indicators such as length, weight and head circumference.

The diagnostic criteria for MBDP are not unified. There still lacks a consensus on the screening methods of MBDP. The identification and intervention of MBDP risk factors can reduce its incidence. Despite of many studies on the risk factors of MBDP in China and overseas, their research results are not consistent due to regional differences. Therefore, the purpose of this study is to conduct a Meta analysis on the collected literature pertaining to the risk factors of MBDP, in an effort to reduce its incidence and provide a reference for the prevention of MBDP.

## 2 Materials and methods

### 2.1 Literature search

By using "zao chan er dai xie xing gu bing", "wei xian yin su", "xiang guan yin su" "ying xiang yin su" as Chinese search terms; "metabolic bone disease of prematurity", "risk factor" and

"risk" as English search terms. We systematically searched the China Biomedical Literature Service System, China National Knowledge Infrastructure, Wanfang Data, Weipu Periodical Database, PubMed, Web of Science, Cochrane Library, and Embase databases from inception to June 18, 2021 with no restrictions on language, population or publication year. In addition, we manually searched the reference lists of the included studies to identify additional relevant literature. We also searched the Chinese Clinical Registry and the American Clinical Registry to obtain more unpublished related literature.

## 2.2 Literature inclusion and exclusion criteria

Inclusion criteria: (1) Original research on the risk factors of MBDP from the beginning of the establishment of the above databases until June 18, 2021; (2) The type of study design is a case-control study or the study population is divided into case group and control group, and the current situation or retrospective study comparing the two groups of exposure factors; (3) The diagnostic criteria of the disease and the definition and quantification of exposure factors are basically the same; (4) The literature directly or indirectly provides the OR (95% *CI*) of the exposure factors.(5)If the study involving the same population has been published more than once, the study with a larger sample size or with the most recent data was selected; Exclusion criteria: (1)Duplicate publications; (2)Reviews, systematic reviews, animal experiments; (3) Inconsistencies in research content; (4) Inconsistent experimental methods; (5) Unable to obtain the full text; (6) The outcome indicators do not match or are missing.

## 2.3 Document data extraction

The literature was screened and the data were extracted independently by two reviewers and cross-checked. If inconsistencies were encountered, they were resolved by discussion. If necessary, the decision was made by a third party. Any missing information was supplemented by contact with the author. The process of literature screening was as follows: exclude the duplicate studies; read the titles and abstracts to exclude irrelevant articles, and read the full text to identify the included studies. The literature data extraction includes the name of the first author, publication year, study area, study time, study design type, number of case groups and control groups, exposure factors and OR (95% *CI*). After the extraction was completed, a third person would check the results of the extracted data, and deal with the differences between the data through group discussion and consultation with professional statisticians.

## 2.4 Literature quality evaluation

Two independently evaluated the quality of the literature, and finally summarized them. Case-control studies and retrospective cohort studies refer to the Newcastle-Ottawa Scale (NOS) to evaluate the quality of the literature. When the score is greater than or equal to 7 points (out of 9 points), it can be regarded as high-quality literature. Cross-sectional studies uses the evaluation criteria recommended by the American Health Care Quality and Research Institute (AHRQ) (full score of 11) to evaluate the literature, and the score $\geq$ 8 is classified as high-quality literature [8].

## 2.5 Statistical analysis

Used Excel 2013 software to establish a database and verified it. The forest map was produced using Review Manager (RevMan) 5.3 software, and the rest of the statistical analysis (such as funnel graph production, publication bias detection) was carried out using Stata 14.1 software. The effect size is the OR value of the influencing factors of MBDP and its 95% *CI*. The $I^2$ value

and the Cochran Q test were used to test the heterogeneity. If $I^2 > 50\%$ or $P < 0.1$, it indicates that the results are heterogeneous, and the random effects model (REM) analysis is used; Otherwise, the fixed effects model (FEM) is used. By comparing the differences in the combined values of different effect models, the sensitivity of the research results is analyzed. Used funnel plots and Egger's linear regression to assess potential publication bias.

## 3 Results

### 3.1 The basic situation of the included literature

A total of 467 documents were obtained from the preliminary search. According to the inclusion and exclusion criteria, 15 articles were finally included [9–23], including 5 Chinese articles [9–13] and 8 English articles [14–19, 22, 23], 2 Spanish article [20, 21]. The literature screening process and results are shown in Fig 1. The included literature research sites are from 7 countries (China, Turkey, Spain, United States, Mexico, Sweden, Canada); the total sample size is 3492 cases, including 1,435 cases in the case group and 2,057 cases in the control group. The basic information of the included literature is shown in Table 1. Thirteen case-control studies are high-quality studies, one comparative cross-sectional study is high-quality research, and one retrospective cohort study is high-quality research. The quality evaluation table is shown in Table 2 (NO.1-NO.11, NO.13-NO.15) and Table 3 (NO.12).

### 3.2 Meta analysis results

According to the risk factors of metabolic bone disease of prematurity involved in the included literature, 8 related factors were selected for analysis. The results of heterogeneity analysis showed that the two factors of septicemia and parenteral nutrition time are less heterogeneous among different studies and was analyzed by fixed-effects model. Other factors are more heterogeneous between different studies and are analyzed by random-effects model. The results of Meta analysis showed that the combined OR values of the 8 factors included (birth weight, birth weight <1000g, gestational age, gestational age <32 weeks, Septicemia, parenteral nutrition time, cholestasis, intrauterine growth retardation) have all Statistically significant ($P$<0.05). The analysis results are shown in Table 4 and Figs 2–9.

### 3.3 Sensitivity analysis and publication bias

By comparing the results of the fixed-effects model and the random-effects model, the sensitivity analysis showed that the combined effect values of the two models for each risk factor did not differ significantly, as shown in Table 5. The funnel chart shows that the funnel chart of each risk factor in this study is symmetrical, indicating that there is no publication bias, see Figs 10–17. Egger's test results showed that $P$>0.05, indicating that there is no publication bias, see Table 5.

## 4 Discussion

Metabolic bone disease of prematurity is affected by many factors. The vast majority of children have no obvious clinical symptoms. Its diagnosis mainly depends on early clinical screening and monitoring. Therefore, it is a challenge to choose the best screening method at the appropriate time [5]. Clarifying the high risk factors of MBDP can serve the prevention of the disease. This study collected related studies and performed meta analysis. The results showed that high-risk factors for MBDP include birth weight <1000g, gestational age <32 weeks, septicemia, parenteral nutrition time, cholestasis, and intrauterine growth retardation, while birth weight and gestational age are its protective factors.

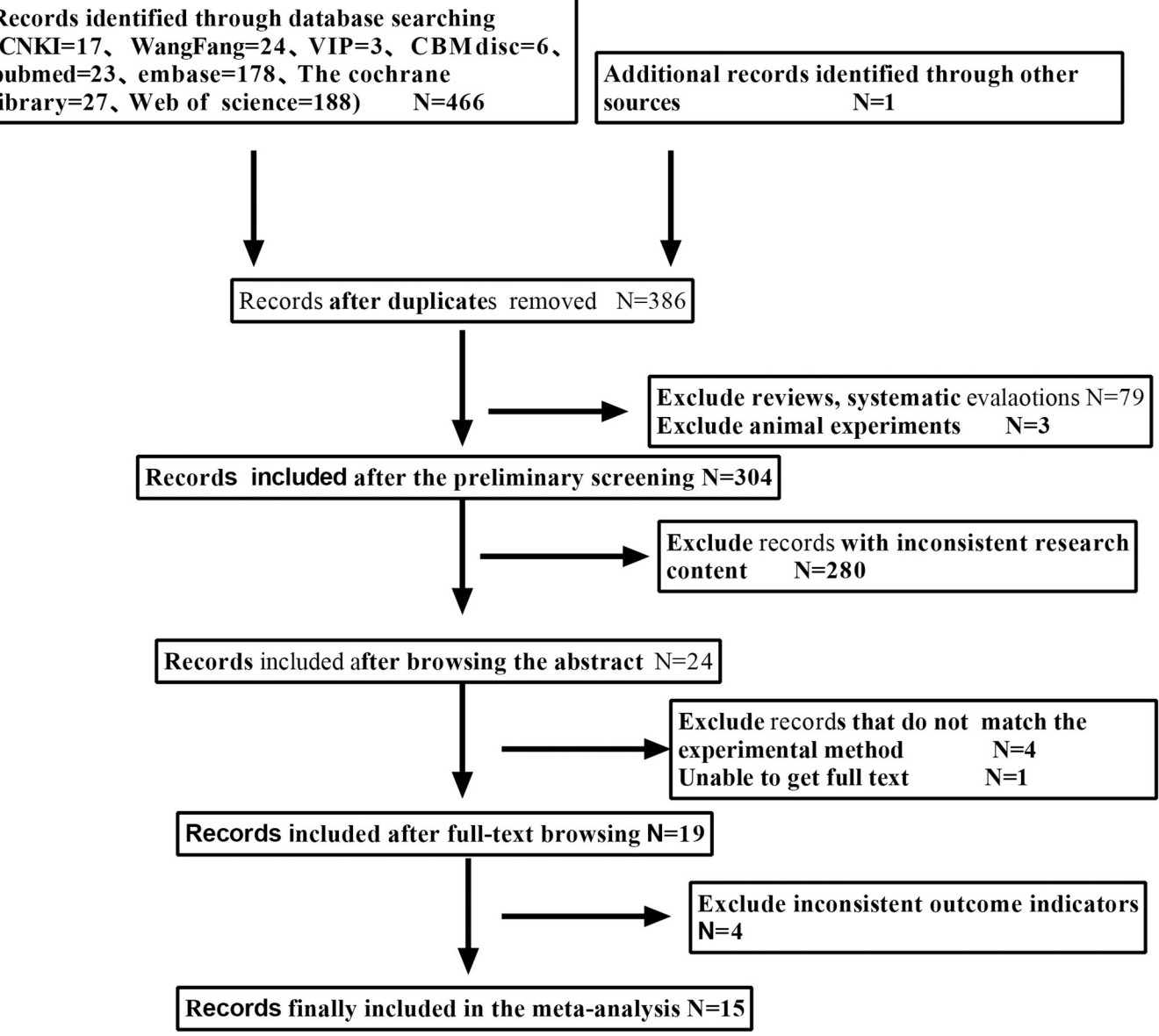

**Fig 1. Document screening flow chart.**

The results of this study show that gestational age <32 weeks is a risk factor for MBDP (OR = 2.73), while gestational age is a protective factor for MBDP (OR = 0.57); It indicates that the smaller the gestational age, the higher the risk of MBDP, and as the gestational age increases, the risk of MBDP decreases. It is mainly related to the following two reasons. The first is the lack of fetal mineral reserves due to premature delivery. At 25 to 40 weeks of gestation, the total amount of calcium and phosphorus accumulated in the fetus accounts for 80% of the total amount of calcium and phosphorus in the body [4, 24, 25]. The average deposition rate of calcium and phosphorus during this period was 100-120mg/kg/day and 50-65mg/kg/day, which can provide 20g calcium and 10g phosphorus reserves for newborns. If premature birth occurs during this period, the newborn may miss the optimal stage of obtaining calcium and phosphorus reserves [26]. Second, during the hospitalization of premature infants, due to

**Table 1. Basic information of included literature.**

| serial number | literature | Study area | Research time | Type of Study | Number of case group | Number of control group | Risk factors |
|---|---|---|---|---|---|---|---|
| NO.1 | Xiaori He 2021 [9] | China | September 1,2013-August 31,2016 | Case control | 108 | 396 | ①②③④ |
| NO.2 | Meixi Wang 2021 [10] | Chengde, China | October,2015—July,2019 | Case control | 101 | 125 | ⑤⑥⑦⑧ |
| NO.3 | Wei Wang 2020 [11] | Xi'an, China | January,2017—March,2019 | Case control | 226 | 226 | ⑤⑥⑨⑩⑪⑫□ |
| NO.4 | Jiaxin Xu 2019 [12] | Qingdao, China | January,2016—December,2017 | Case control | 58 | 116 | ⑥⑨⑬⑭⑮⑯⑰ |
| NO.5 | Meixi Wang 2019 [13] | Chengde, China | October,2015—July,2018 | Case control | 149 | 148 | ⑤⑥⑩ |
| N0.6 | Mehmet Mutlu 2021 [14] | Turkey | 2015year—2018year | Case control | 81 | 63 | ⑱ |
| NO.7 | Hui Zhang 2021 [15] | Beijing China | January,2014—December,2019 | Case control | 73 | 69 | ⑯⑲⑳ |
| NO.8 | Wenwen Chen 2021 [16] | Zhangzhou, China | June,2016—May,2020 | Case control | 52 | 104 | ⑩□⑨⑪④⑫ |
| NO.9 | Alejandro Avila-Alvarez 2020 [17] | Spain | January1,2015—July31,2020 | Case control | 27 | 191 | ⑥ |
| NO.10 | Wenhao Chen 2018 [18] | Fujian China | January1,2011—November30,2015 | Case control | 16 | 32 | ①⑨□ |
| NO.11 | Supamit Ukarapong 2017 [19] | United States | January,2013—April,2014 | Case control | 40 | 36 | ⑪ |
| NO.12 | Rios-Moreno 2016 [20] | Mexico | January,2011—January,2012 | Comparative cross-sectional study | 58 | 62 | ④□□□□ |
| NO.13 | Alicia Montaner Ramón 2017 [21] | Spain | January,2012—December,2014 | Case control | 21 | 118 | ⑩ |
| NO.14 | Högberg Ulf 2018 [22] | Sweden | 1997 year—2014 year | Case control | 316 | 188 | ①⑲□□ |
| NO.15 | Ebtihal Ali 2018 [23] | Canada | October,2007—June, 2012 | Cohort retrospective study | 109 | 183 | ⑤⑧□□□ |

Note: ① Gestational age <32 weeks ② Hypocalcemia ③ Extrauterine growth retardation at discharge ④ Septicemia ⑤ Gestational age ⑥ Birth weight ⑦ Caffeine treatment duration ⑧ Caffeine involved dose ⑨ Parenteral nutrition time ⑩ Intrauterine growth retardation ⑪ Cholestasis ⑫ Diuretic application ⑬ Small for gestational age ⑭ Hospital time ⑮ Ventilator support time ⑯ Breast milk ⑰ Starting enteral feeding time ⑱ Antiepileptic drug use ⑲ Male ⑳ Initial serum alkaline phosphatase ㉑ Birth weight<1000g ㉒ VitD supplementation after 14 days of age ㉓ Moderate to severe BPD ㉔ Sedation time ㉕ Duration of corticosteroid application ㉖ Maternal overweight/obesity ㉗ vitamin D deficiency ㉘ Steroid cumulative dose ㉙ Average biweekly Birth Weight.

their small gestational age or the need for ventilator-assisted ventilation, they are in immobilization and lack motor stimulation, so there may be a risk of bone mineralization defects [27, 28]. Skeletal demineralization in the neonatal period may be the result of inactivity due to nervous system, neuromuscular or systemic metabolic diseases [29].

Birth weight <1000g is a risk factor for MBDP (OR = 6.62), while birth weight is a protective factor for MBDP (OR = 0.44); it indicates that the lower the birth weight, the higher the risk of MBDP, and as the birth weight increases, the risk of MBDP is reduced accordingly. It is mainly related to the following two reasons. The first is related to premature birth. Fetal bone mineral accumulation is mainly in the third trimester, and premature babies will miss the main opportunity for mineral accumulation. After birth, it is difficult to maintain a comparable mineral intake [30]. Second, low birth weight may be related to placental insufficiency, and any situation that impairs placental function and therefore impairs nutrient transfer may increase the risk of MBDP, which will lead to a decrease in mineral transfer [4].

**Table 2. Quality evaluation form (NOS scale).**

| serial number | literature | Type of Research | selection | comparability | outcome | Total score |
|---|---|---|---|---|---|---|
| NO.1 | Xiaori He 2021 [9] | Case control | ☆☆☆ | ☆☆ | ☆☆☆ | 8 point |
| NO.2 | Meixi Wang 2021 [10] | Case control | ☆☆☆ | ☆☆ | ☆☆☆ | 8 point |
| NO.3 | Wei Wang 2020 [11] | Case control | ☆☆☆ | ☆☆ | ☆☆☆ | 8 point |
| NO.4 | Jiaxin Xu 2019 [12] | Case control | ☆☆☆ | ☆☆ | ☆☆☆ | 8 point |
| NO.5 | Meixi Wang 2019 [13] | Case control | ☆☆☆ | ☆☆ | ☆☆☆ | 8 point |
| N0.6 | Mehmet Mutlu 2021 [14] | Case control | ☆☆☆ | ☆☆ | ☆☆☆ | 8 point |
| NO.7 | Hui Zhang 2021 [15] | Case control | ☆☆☆ | ☆☆ | ☆☆☆ | 8 point |
| NO.8 | Wenwen Chen 2021 [16] | Case control | ☆☆☆ | ☆☆ | ☆☆ | 7 point |
| NO.9 | Alejandro Avila-Alvarez 2020 [17] | Case control | ☆☆☆ | ☆☆ | ☆☆☆ | 8 point |
| NO.10 | Wenhao Chen 2018 [18] | Case control | ☆☆☆ | ☆☆ | ☆☆☆ | 8 point |
| NO.11 | Supamit Ukarapong 2017 [19] | Case control | ☆☆☆ | ☆☆ | ☆☆☆ | 8 point |
| NO.13 | Alicia Montaner Ramón 2017 [21] | Case control | ☆☆☆ | ☆☆ | ☆☆☆ | 8 point |
| NO.14 | Högberg Ulf 2018 [22] | Case control | ☆☆☆ | ☆☆ | ☆☆☆ | 8 point |
| NO.15 | Ebtihal Ali 2018 [23] | Cohort retrospective study | ☆☆ | ☆☆ | ☆☆☆ | 7 point |

The results of this study show that septicemia is a risk factor for MBDP (OR = 2.53), and septicemia is one of the common causes of morbidity and death in preterm infants [31]. Jensen EA et al. found that infants with sepsis and bronchopulmonary dysplasia confirmed by blood culture were associated with an increased probability of MBDP [32]. It is mainly caused by the interaction between the immune system and the skeletal system [33]. Lipopolysaccharide exposure may cause bone loss [34], which may be due to the activation of B cells and T cells that may regulate bone resorption [33]. In addition, the treatment of sepsis will also prolong the use of parenteral nutrition and [32] increase the risk of MBDP. We should adopt strict hygiene procedures, minimize invasive interventions, and supplement probiotics in the pre-term birth of exclusive breastfeeding [35] to prevent sepsis, thereby reducing the incidence of MBDP.

Parenteral nutrition time is a risk factor for MBDP (OR = 4.04). Premature infants often cannot eat in the early postpartum period or cannot achieve total enteral nutrition in the short term, so long-term parenteral nutrition is required. However, parenteral nutrition formulations often fail to provide sufficient or usable mineral supply due to various factors, including the lack of corresponding mineral formulations, poor solubility of minerals, mutual antagonism of nutrients, and the influence of pH, etc [36]. Therefore, the deposition of calcium and phosphorus in the early postpartum period of preterm infants cannot meet the requirements of intrauterine bone growth rate [26, 37]. In addition, there are reports that aluminum contamination of parenteral nutrition can cause MBDP [38, 39]. Aluminum contamination of parenteral nutrition can lead to excessive deposition of aluminum on the surface of bone mineralization, which affects the activity of osteoblasts and hinders bone formation, ultimately leading to osteomalacia. Since aluminum is released during the sterilization of glass bottles [39], it is difficult to avoid aluminum contamination of parenteral nutrition. Every effort should be made to speed up the transition of preterm infants receiving parenteral nutrition to enteral feeding. The fortified formula has the right mineral ratio, balanced nutrients, and the

**Table 3. AHRQ cross-sectional study evaluation standard score.**

| serial number | literature | Type of Research | Score situation | Total score |
|---|---|---|---|---|
| NO.12 | Rios-Moreno 2016 [20] | Comparative cross-sectional study | Articles 5, 9, and 11 are unclear | 8 point |

**Table 4. Results of heterogeneity test and Meta analysis of MBDP risk factors.**

| Research factors | Literature source | Heterogeneity test | | | Effect model | merge OR (95%CI) | merge P value |
|---|---|---|---|---|---|---|---|
| | | Q | I² | P | | | |
| Birth weight | [10, 12, 13, 17] | 20.60 | 85 | 0.0001 | Random effect | 0.44 (0.21–0.90) | P = 0.02 |
| Birth weight<1000g | [11, 16, 20] | 10.58 | 81 | 0.005 | Random effect | 6.62 (2.28–19.25) | P = 0.0005 |
| Gestational age | [10, 13, 23] | 4.17 | 52 | 0.12 | Random effect | 0.57 (0.44–0.73) | P<0.00001 |
| Gestational age<32 week | [9, 11, 18, 22] | 94.32 | 97 | <0.00001 | Random effect | 2.73 (1.07–6.95) | P = 0.03 |
| Septicemia | [9, 16, 20] | 1.79 | 0 | 0.41 | Fixed effect | 2.53 (1.69–3.79) | P<0.00001 |
| Parenteral nutrition time | [11, 12, 16, 18] | 32.68 | 91 | <0.00001 | Random effect | 4.04 (1.72–9.49) | P = 0.001 |
| Cholestasis | [11, 16, 19] | 5.61 | 64 | 0.06 | Random effect | 3.50 (1.49–8.23) | P = 0.004 |
| Intrauterine growth retardation | [11, 13, 16, 21] | 0.64 | 0 | 0.89 | Fixed effect | 6.89 (3.81–12.44) | P<0.00001 |

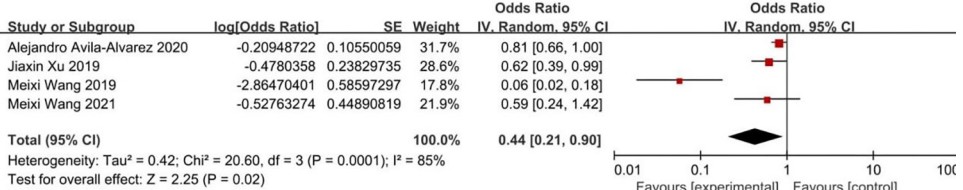

**Fig 2. Forest diagram of the relationship between birth weight and the incidence of MBDP.**

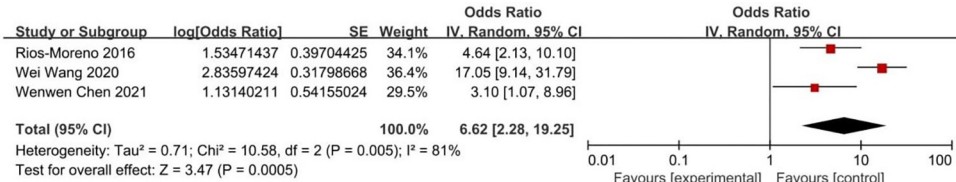

**Fig 3. Forest diagram of the relationship between birth weight <1000g and the incidence of MBDP.**

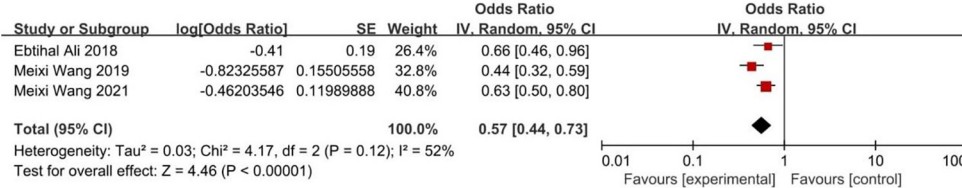

**Fig 4. Forest diagram of the relationship between gestational age and the incidence of MBDP.**

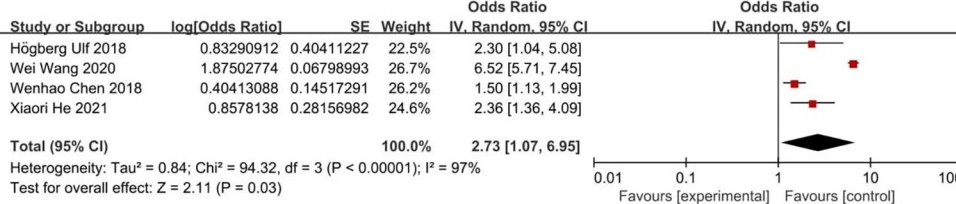

**Fig 5. Forest diagram of the relationship between gestational age <32 weeks and the incidence of MBDP.**

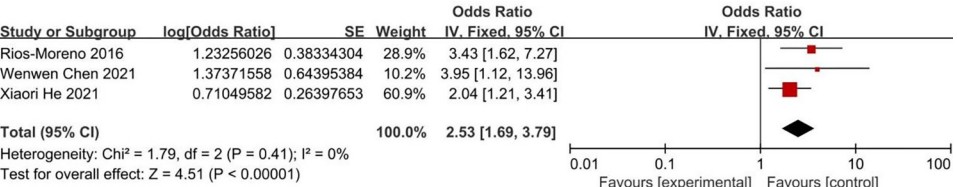

**Fig 6. Forest diagram of the relationship between septicemia and the incidence of MBDP.**

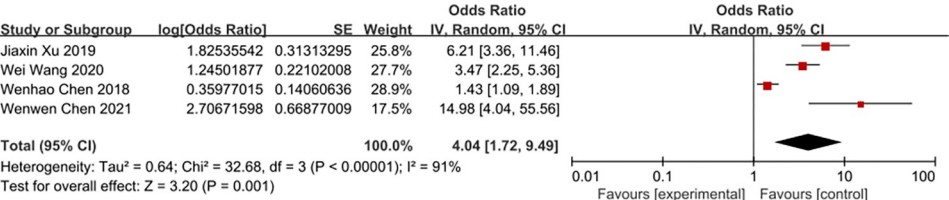

**Fig 7. Forest diagram of the relationship between parenteral nutrition time and the incidence of MBDP.**

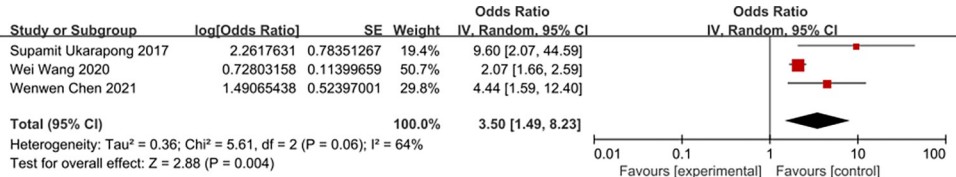

**Fig 8. Forest diagram of the relationship between cholestasis and the incidence of MBDP.**

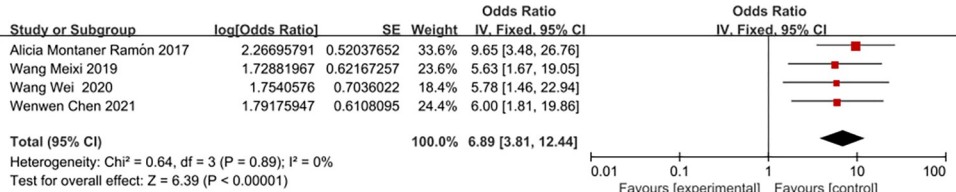

**Fig 9. Forest diagram of the relationship between intrauterine growth retardation and the incidence of MBDP.**

**Table 5. Sensitivity analysis and publication bias test.**

| Research factors | Sensitivity analysis | | Egger's test | |
|---|---|---|---|---|
| | Fixed effects model OR(95%*CI*) | Random effects model OR(95%*CI*) | T value | P value |
| Birth weight | 0.72(0.60–0.86) | 0.44(0.21–0.90) | -1.97 | 0.187 |
| Birth weight<1000g | 8.33(5.35–12.96) | 6.62(2.28–19.25) | -2.03 | 0.291 |
| Gestational age | 0.57(0.48–0.67) | 0.57(0.44–0.73) | -0.14 | 0.909 |
| Estational age <32 weeks | 4.76(4.23–5.35) | 2.73(1.07–6.95) | -1.34 | 0.312 |
| Septicemia | 2.53(1.69–3.79) | 2.53(1.69–3.79) | 1.74 | 0.333 |
| Parenteral nutrition time | 2.27(1.83–2.81) | 4.04(1.72–9.49) | 3.21 | 0.085 |
| Cholestasis | 2.21(1.78–2.74) | 3.50(1.49–8.23) | 9.50 | 0.067 |
| Intrauterine growth retardation | 6.89(3.81–12.44) | 6.89(3.81–12.44) | -2.58 | 0.123 |

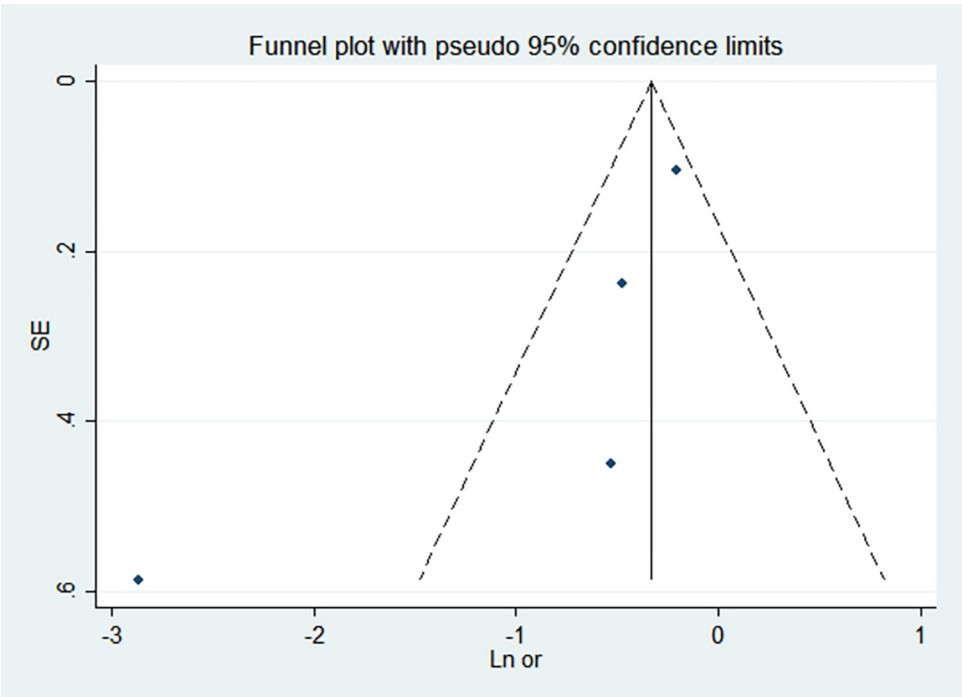

**Fig 10. Funnel chart of birth weight.**

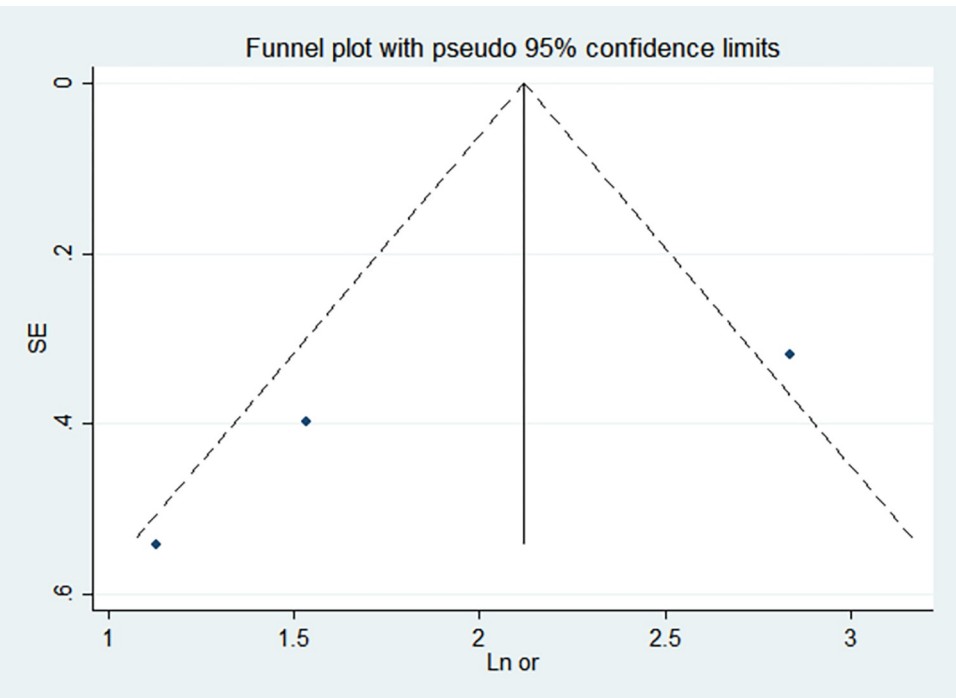

**Fig 11. Funnel chart of birth weight <1000g.**

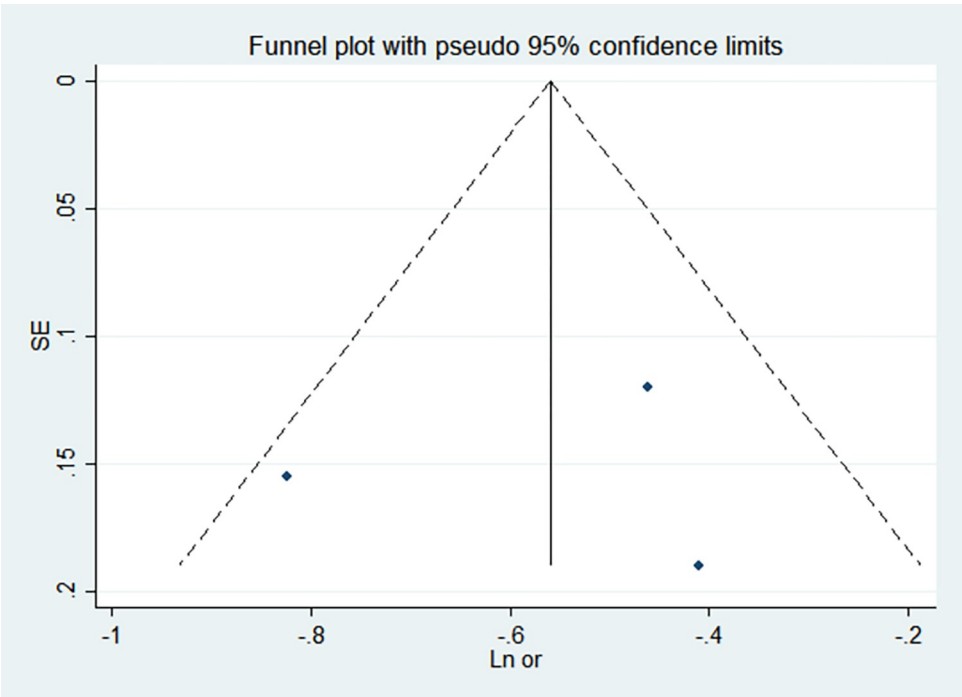

**Fig 12. Funnel chart of gestational age.**

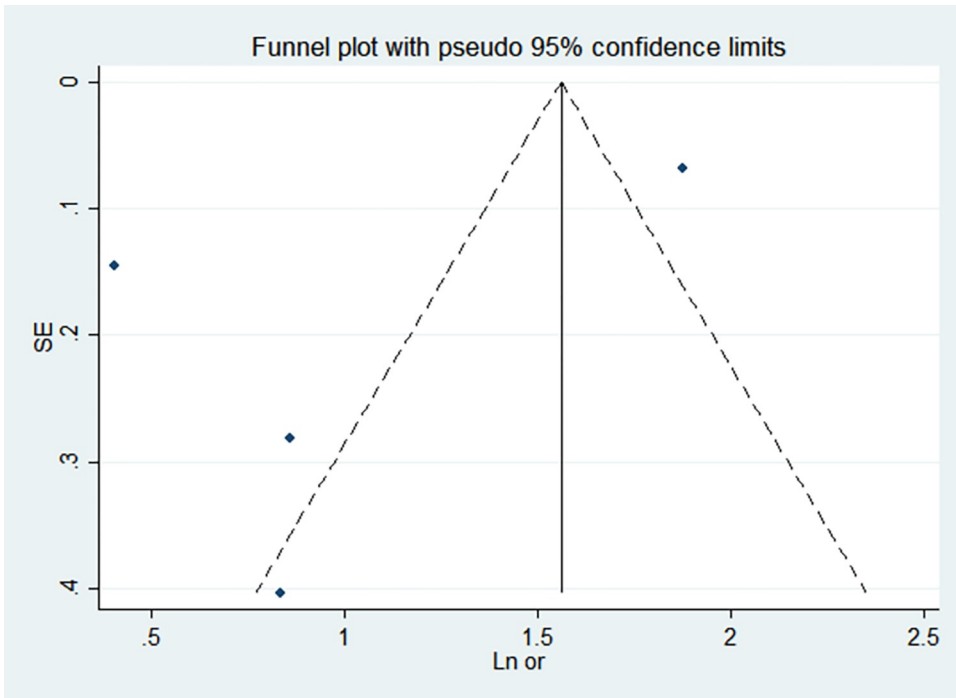

**Fig 13. Funnel chart of gestational age <32 weeks.**

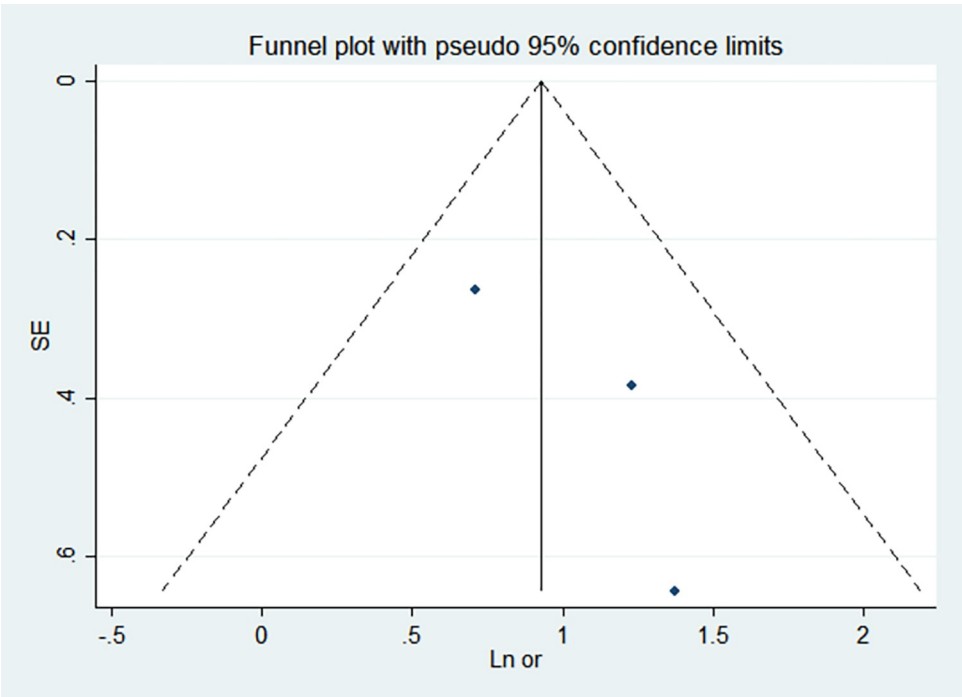

**Fig 14. Funnel chart of septicemia.**

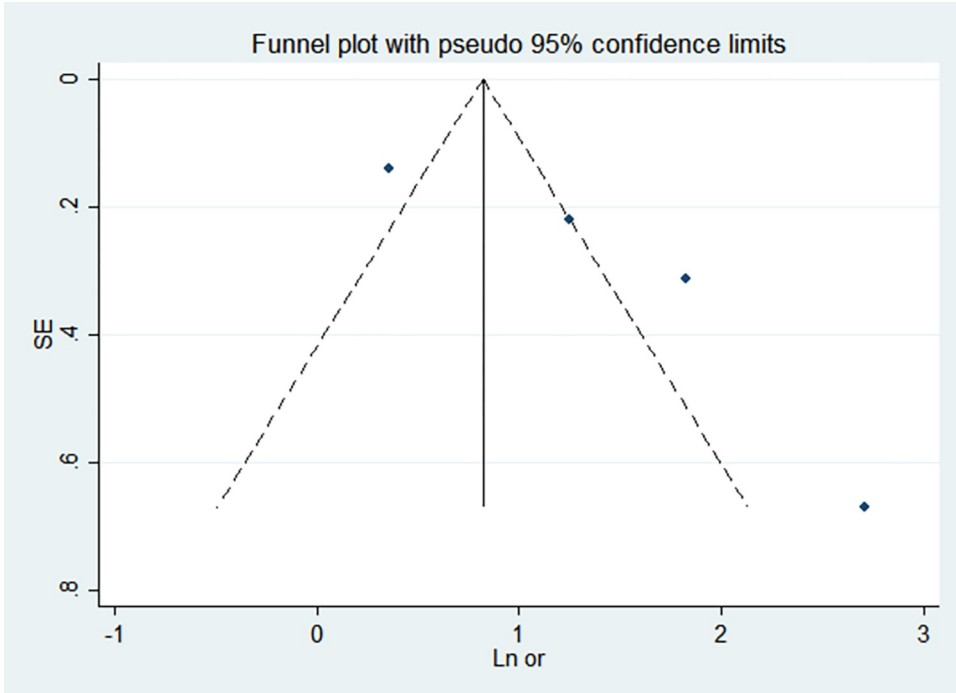

**Fig 15. Funnel chart of parenteral nutrition time.**

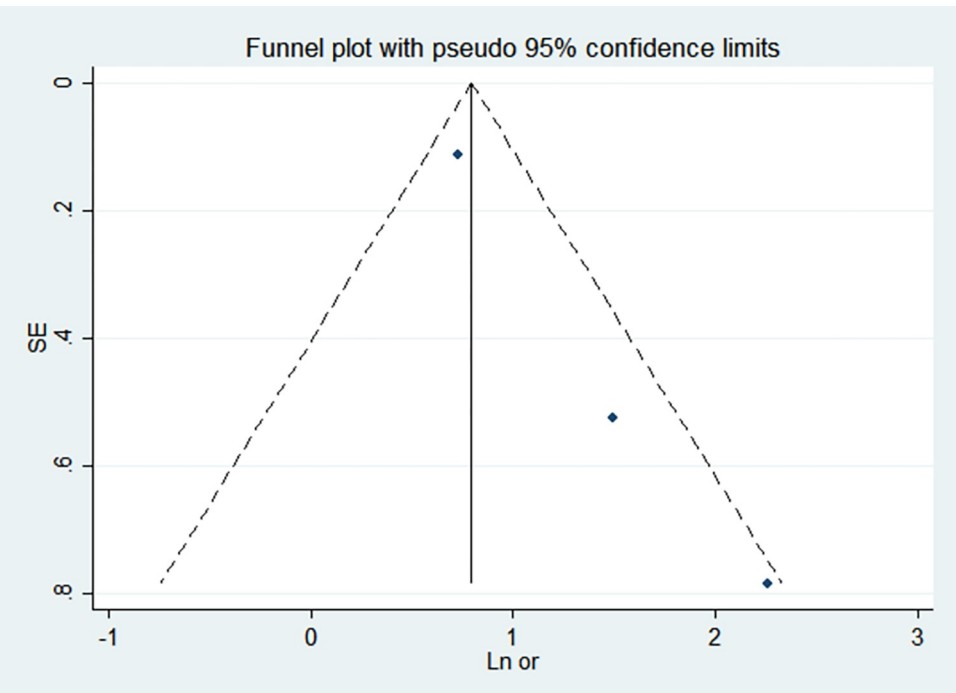

**Fig 16. Funnel chart of cholestasis.**

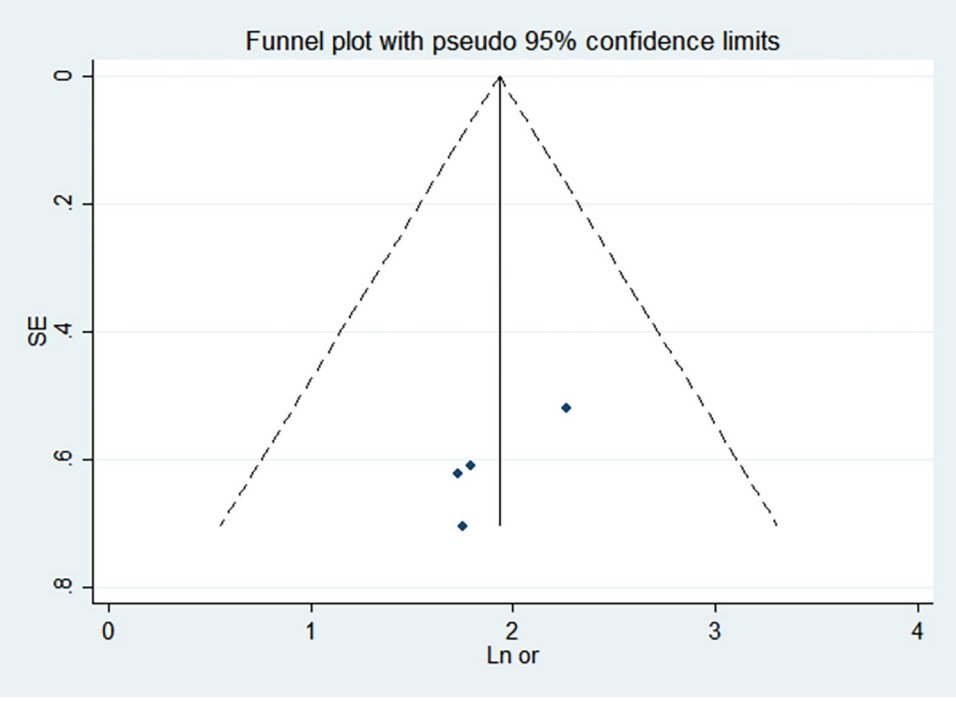

**Fig 17. Funnel chart of intrauterine growth retardation.**

calcium-phosphorus ratio is similar to breast milk. Therefore, the enhanced formula can provide the best calcium and phosphorus deposition rate during intrauterine growth. For example, the gastrointestinal absorption rate of phosphorus can reach more than 90% during enteral feeding [40]. Enteral nutrition has unparalleled advantages in ensuring the absorption efficiency of minerals such as calcium and phosphorus in the early postpartum period of premature infants.

Cholestasis is a risk factor for MBDP (OR = 3.5). There are two main reasons. First, cholestasis is related to the reduction of vitamin D absorption. Premature infants tend to have low serum 25-hydroxyvitamin D levels. This is especially true in premature babies born with a gestational age of <32 weeks [41, 42]. Second, cholestasis can increase bilirubin, bile acid, litho-cholic acid, etc. Bilirubin and bile acid have a negative effect on the function of osteoblasts. Ruiz-Gaspà S et al. reported that the bilirubin and serum of patients with jaundice had harmful effects on the proliferation and mineralization of primary human osteoblasts and SAOS-2 human osteosarcoma cells. In addition, they also found that lithocholic acid affects the original. The viability of the generation of human osteoblasts [43]. Lithocholic acid can interfere with the absorption of vitamin D as an analog of vitamin D [44].

Intrauterine growth retardation (IUGR) is an independent risk factor for MBDP (OR = 6.89). Vitamin D3 deficiency can lead to poor placental implantation, and changes in trophoblasts can induce IUGR [45–47]. It is possible that the association between IUGR and maternal vitamin D3 deficiency leads to decreased intrauterine bone calcification. Chronic damage to the placenta can also directly affect phosphorus transport, and can also lead to blocked bone mineralization [48].

There are some limitations in this study. Firstly, some risk factors rarely reported are not included, which undermines the comprehensive coverage of the risk factors of MBDP. Secondly, the reviewed literature vary in their methodological differences, ranging from case-control studies, current status surveys, to retrospective cohort studies, leading to high heterogeneity of some factors. Lastly, the reviewed literature in this study are only limited to Chinese, English, and Spanish. The absense of publications in other language may cause language bias.

In summary, this meta-analysis evaluates the risk factors and their correlation with MBDP. Among them, birth weight <1000g, gestational age <32 weeks, septicemia, parenteral nutrition time, cholestasis, intrauterine growth retardation are high-risk factors for MBDP, while birth weight and gestational age are its protective factors. Therefore, to prevent the occurrence of MBDP, it is necessary to strengthen the perinatal health care of pregnant and lying-in women, reduce their underlying health conditions, and avoid infants to be born at a too small gestational age and with low birth weight. At the same time, strict hygienic procedures should be adopted to minimize invasive interventions to reduce the occurrence of sepsis, and shorten the time of parenteral nutrition as much as possible.

## Supporting information

**S1 File. PRISMA_2020_checklist.**
(DOCX)

**S2 File. Search strategy.**
(DOCX)

**S3 File. Data extracted from included studies.**
(DOC)

## Author Contributions

**Conceptualization:** Jie Wang, Qian Zhao.

**Data curation:** Jie Wang, Qian Zhao, Baochang Chen, Weichen Yan.

**Formal analysis:** Jie Wang, Qian Zhao, Jingfei Sun, Weichen Yan.

**Investigation:** Jie Wang, Baochang Chen, Jingfei Sun, Weichen Yan.

**Methodology:** Jie Wang, Changjun Ren.

**Resources:** Jie Wang, Jinfeng Meng, Shangbin Li, Changjun Ren.

**Software:** Jie Wang, Qian Zhao, Changjun Ren.

**Supervision:** Jie Wang, Baochang Chen, Jiayu Huang, Jinfeng Meng, Ling Hao.

**Validation:** Jie Wang, Jingfei Sun, Jiayu Huang, Shangbin Li, Ling Hao.

**Visualization:** Jie Wang, Jiayu Huang, Shangbin Li, Ling Hao.

**Writing – original draft:** Jie Wang, Qian Zhao, Changjun Ren.

**Writing – review & editing:** Jie Wang, Qian Zhao, Jinfeng Meng, Changjun Ren, Ling Hao.

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
