## [Decision Letter · Decision Letter 0]

26 Apr 2022

PONE-D-21-28993Risk factors for metabolic bone disease of prematurity：A meta-analysisPLOS ONE

Dear Dr. Ren,

Thank you for submitting your manuscript to PLOS ONE. After careful consideration, we feel that it has merit but does not fully meet PLOS ONE’s publication criteria as it currently stands. Therefore, we invite you to submit a revised version of the manuscript that addresses the points raised during the review process.

The manuscript has been reviewed by 2 experts in the field.  While the findings are interesting, there remain some minor concerns with the manuscript. The author is invited to revise and resubmit the manuscript.  The authors should respond to each of the comments. 

We look forward to receiving your revised manuscript.

Kind regards,

Dengshun Miao

Academic Editor

PLOS ONE

Journal Requirements:

2. Please provide the full electronic search strategy for at least one database, including any limits used, such that it could be repeated"". No need to ping with follow up.

5. We note that you have referenced (ie. Bewick et al. [5]) which has currently not yet been accepted for publication. Please remove this from your References and amend this to state in the body of your manuscript: (ie “Bewick et al. [Unpublished]”) as detailed online in our guide for authors

Reviewers' comments:

Reviewer's Responses to Questions

**Comments to the Author**

1. Is the manuscript technically sound, and do the data support the conclusions?

Reviewer #1: Yes

Reviewer #2: Yes

2. Has the statistical analysis been performed appropriately and rigorously? 

Reviewer #1: I Don't Know

Reviewer #2: N/A

3. Have the authors made all data underlying the findings in their manuscript fully available?

Reviewer #1: Yes

Reviewer #2: Yes

4. Is the manuscript presented in an intelligible fashion and written in standard English?

Reviewer #1: Yes

Reviewer #2: Yes

5. Review Comments to the Author

Reviewer #1: In this meta-analysis, they investigated risk factors for metabolic bone disease of prematurity (MBDP). The paper seemed to be useful and well prepared.

However, MBDP is not well known as the disease entity in most countries, and probably specific to the limited countries, including China. Therefore, the paper is difficult to understand for most general readers.

Definition, symptom, clinical and laboratory findings, diagnosis, treatment and prognosis of MBDP　should be clearly and concisely explained in Introduction.

Reviewer #2: Wang et al. investigated the risk factors for metabolic bone disease of prematurity (MBDP) by using a meta analysis to re-analyze the cases with MBDP in the literature, including 1,435 cases in the experimental group and 2,057 cases in the control group. They claimed that birth weight (below 1000g), gestational age (below 32 weeks), septicemia, parenteral nutrition time, cholestasis, and intrauterine growth retardation may increase the risk of MBDP. These findings may contribute to the understanding of prevention of MBDP. I would suggest the authors use their own cohort to further confirm their conclusions.

6. PLOS authors have the option to publish the peer review history of their article (what does this mean?). If published, this will include your full peer review and any attached files.

Reviewer #1: No

Reviewer #2: No

---

## [Author Response · Author response to Decision Letter 0]

8 May 2022

Dear Editors and Reviewers:

Thank you for your letter and for the Editors and reviewers’ comments concerning our manuscript entitled “Risk factors for metabolic bone disease of prematurity：A meta-analysis”(ID: PONE-D-21-28993R1). Those comments are all valuable and very helpful for revising and improving our paper, as well as the important guiding significance to our researches. 

We have studied comments carefully and have made correction which we hope meet with approval. Revised portion are marked in red in the paper. The main corrections in the paper and the responds to the reviewer’s comments are as flowing: 

Responds to the editor’s comments: 

1.Response to comment 1: Please ensure that your manuscript meets PLOS ONE's style requirements, including those for file naming. Response: It is really true as Editor’s suggest, We have re-read the article, and the style of the article and title meets the requirements of Plos one.If there is something that needs to be corrected or amended, please contact me in time and I will amend it on time.

2.Response to comment 2: Please provide the full electronic search strategy for at least one database, including any limits used, such that it could be repeated"".Response: We accept Editor’s good suggestion, we submitted the search method of Pubmed in the supplementary file.It has been uploaded, please check it.

3.Response to comment 3:Thank you for stating the following financial disclosure:The funders had no role in study design, data collection and analysis, decision to publish, or preparation of the manuscript.At this time, please address the following queries:Response:We don’t receive any funding for this study,and in cover letter,We write “The authors received no specific funding for this work.”

4.Response to comment 4:We note that you have stated that you will provide repository information for your data at acceptance. Should your manuscript be accepted for publication, we will hold it until you provide the relevant accession numbers or DOIs necessary to access your data.Response:in cover letter,We write “Data Availability Statement: All relevant data are within the manuscript and its Supporting Information files.”For example,data extracted from included studies and the final 15 articles included in this study.

5.Response to comment 5:We note that you have referenced (ie. Bewick et al. [5]) which has currently not yet been accepted for publication. Please remove this from your References and amend this to state in the body of your manuscript: (ie “Bewick et al. [Unpublished]”) .Response:We have not cited this literature and have checked and updated the references for all of them.We checked and updated all references as required,we added the PMID.

6.Response to comment 6: Please include captions for your Supporting Information files at the end of your manuscript, and update any in-text citations to match accordingly. Please see our Supporting Information guidelines for more information.Response: We accept Editor’s suggestion, we have added the title in the supplementary file to the end of the article.At present, we have four Supporting Information.

Responds to the Reviewer’s comments:

1.Reviewer #1: In this meta-analysis, they investigated risk factors for metabolic bone disease of prematurity (MBDP). The paper seemed to be useful and well prepared.However, MBDP is not well known as the disease entity in most countries, and probably specific to the limited countries, including China. Therefore, the paper is difficult to understand for most general readers.Definition, symptom, clinical and laboratory findings, diagnosis, treatment and prognosis of MBDP　should be clearly and concisely explained Introduction.Response:Based on the first reviewer's suggestion, this revision, in the introduction, explains the definition, symptoms, clinical and laboratory findings, diagnosis, treatment, and prognosis of MBDP.See pages 3 to 4.

2.Reviewer #2: Wang et al. investigated the risk factors for metabolic bone disease of prematurity (MBDP) by using a meta analysis to re-analyze the cases with MBDP in the literature, including 1,435 cases in the experimental group and 2,057 cases in the control group. They claimed that birth weight (below 1000g), gestational age (below 32 weeks), septicemia, parenteral nutrition time, cholestasis, and intrauterine growth retardation may increase the risk of MBDP. These findings may contribute to the understanding of prevention of MBDP. I would suggest the authors use their own cohort to further confirm their conclusions.Response:The second reviewer's suggestion used our ownown study to support our conclusions, and to date, our own hospital does not have its own cohort study, but in the multicenter study published in June 2021 in Chin J Contemp Pediatr, "Risk factors for metabolic bone disease of prematurity in very/extremely low birth weight infants: a multicenter investigation in China," doi: 10.7499/j.issn.1008-8830.2012055. One of the participating institutions was Hebei Children's Hospital ( Hebei Fifth General Hospital ), one of the hospitals in Hebei Province. However, we believe that in the near future we will have our own cohort study to explore the risk factors of MBDP in The First Hospital of Hebei Medical University, and we believe that there will be research results in the future.

3.Response to comment2. Has the statistical analysis been performed appropriately and rigorously?Reviewer #1: I Don't Know Reviewer #2: N/A;

Response:The statistical analysis of this thesis strictly adhered to the statistical operational procedures. The inclusion and exclusion criteria were developed in detail, and the quality of the literature was carefully evaluated.Used Excel 2013 software to establish a database and verified it. The forest map was produced using Review Manager (RevMan) 5.3 software, and the rest of the statistical analysis (such as funnel graph production, publication bias detection) was carried out using Stata 14.1 software. The effect size is the OR value of the influencing factors of MBDP and its 95% CI. The I 2 value and the Cochran Q test were used to test the heterogeneity. If I 2> 50% or P <0.1, it indicates that the results are heterogeneous, and the random effects model (REM) analysis is used; Otherwise, the fixed effects model (FEM) is used. By comparing the differences in the combined values of different effect models, the sensitivity of the research results is analyzed. Used funnel plots and Egger's linear regression to assess potential publication bias.In addition,Table 1 Basic information of included literature，correcting some of the incorrect information.Two of the 15 included studies were in Spanish, and the one previously written is corrected.See pages 6 to6.

In this article, J. Wang and Qian Zhao actually contributed equally to this work, and J. Wang and Qian Zhao are the first co-authors. Initially, it was just written wrongly, but now it has been corrected to prevail this time.

 Thank you very much for your careful reading and your valuable comments to us.

We tried our best to improve the manuscript and made some changes in the manuscript. 

These changes will not influence the content and framework of the paper. 

Once again, thank you very much for your comments and suggestions. We appreciate for Ediors and Reviewers’warm work earnestly, and hope that the correction will meet with approval.

Yours sincerely,

Changjun Ren

Corresponding author:

Name: Changjun Ren 

E-mail: 137544907@qq.com

---

## [Decision Letter · Decision Letter 1]

17 May 2022

Risk factors for metabolic bone disease of prematurity：A meta-analysis

PONE-D-21-28993R1

Dear Dr. Ren,

We’re pleased to inform you that your manuscript has been judged scientifically suitable for publication and will be formally accepted for publication once it meets all outstanding technical requirements.

Kind regards,

Dengshun Miao

Academic Editor

PLOS ONE

Additional Editor Comments (optional):

Reviewers' comments:

Reviewer's Responses to Questions

**Comments to the Author**

1. If the authors have adequately addressed your comments raised in a previous round of review and you feel that this manuscript is now acceptable for publication, you may indicate that here to bypass the “Comments to the Author” section, enter your conflict of interest statement in the “Confidential to Editor” section, and submit your "Accept" recommendation.

Reviewer #1: All comments have been addressed

Reviewer #2: All comments have been addressed

2. Is the manuscript technically sound, and do the data support the conclusions?

Reviewer #1: (No Response)

Reviewer #2: Yes

3. Has the statistical analysis been performed appropriately and rigorously? 

Reviewer #1: (No Response)

Reviewer #2: Yes

4. Have the authors made all data underlying the findings in their manuscript fully available?

Reviewer #1: (No Response)

Reviewer #2: Yes

5. Is the manuscript presented in an intelligible fashion and written in standard English?

Reviewer #1: (No Response)

Reviewer #2: Yes

6. Review Comments to the Author

Reviewer #1: (No Response)

Reviewer #2: (No Response)

7. PLOS authors have the option to publish the peer review history of their article (what does this mean?). If published, this will include your full peer review and any attached files.

Reviewer #1: No

Reviewer #2: No

---

## [Editor Report · Acceptance letter]

2 Jun 2022

PONE-D-21-28993R1 

Risk factors for metabolic bone disease of prematurity：A meta-analysis 

Dear Dr. Ren:

I'm pleased to inform you that your manuscript has been deemed suitable for publication in PLOS ONE. Congratulations! Your manuscript is now with our production department. 

Kind regards, 

on behalf of

Dr. Dengshun Miao 

Academic Editor

PLOS ONE